# Policy, Price, and Perception: A Phenomenological Qualitative Study of the Rural Food Environment Among Latina Households

**DOI:** 10.3390/ijerph22121800

**Published:** 2025-11-28

**Authors:** Natalia B. Santos, Thais F. Alves, TinaMaria Fernandez, Chad Abresch

**Affiliations:** 1Department of Health Promotion, University of Nebraska Medical Center, Omaha, NE 68198, USA; cabresch@unmc.edu; 2Department of Health Services Research and Administration, University of Nebraska Medical Center, Omaha, NE 68198, USA; thais.alves@unmc.edu; 3HOPE Esperanza, North Platte, NE 69101, USA; tinamarie02@msn.com

**Keywords:** food security, immigration policy, rural economic development, Latinos, Hispanic, public health

## Abstract

Food insecurity disproportionately affects Hispanic households in the US. This study examines food access perceptions among rural Latinos, acknowledging that food environments are complex systems influenced by factors such as availability, accessibility, affordability, acceptability, and accommodation. This phenomenological qualitative study was conducted with adult Latinas living in Nebraska’s rural areas. Data was collected through participatory mapping, semi-structured interviews guided by the five dimensions of food access, and demographic surveys. Eighteen women participated in in-person interviews, and 68.3% of participants met the criteria for food insecurity. While chain stores were the primary shopping option in rural areas, challenges included limited availability of foods that are culturally relevant and accommodate special dietary needs. Ethnic stores were valued for cultural relevance despite concerns about quality and pricing. Overall, affordability was a significant barrier due to high rural costs, worsened by challenges in navigating nutrition program benefits and documentation status. Research or interventions targeting improvements in rural food security must extend beyond mere store availability, focusing on economic development, policy reform, and enhanced education in assistance programs to address these complex challenges.

## 1. Introduction

Food environments are a complex collection of physical, economic, and political conditions that contribute to chronic disease rates in the United States (US) [1]. Rural Latino communities face unique barriers to food access, including limited language skills and a lack of culturally appropriate foods, in addition to traditional barriers faced by rural communities, such as limited availability of stores with fresh produce [2] or increased costs [2]. In 2023, the prevalence of food insecurity was significantly higher for Hispanic households (21.9%) compared to their White, non-Hispanic counterparts (9.9%) [3], along with higher rates of chronic diseases such as diabetes, hypertension, obesity, or chronic liver disease [2]. Literature regarding food environment studies also predominantly focuses on urban settings [4], limiting insights into the status of food environments for Latinos living in rural areas in the US. Compared to urban areas, rural communities experience higher rates of persistent poverty, lower levels of educational attainment, inadequate transportation, a greater proportion of elderly individuals, and limited access to health services [5,6,7,8]. Between 2000 and 2010, Nebraska also saw an increase of over 77% in its Hispanic and Latino population, mostly in rural areas. Ultimately, food environments directly influence a person’s ability to access food.

This study is drawn from the Social Ecological Model (SEM) [9]. By exploring topics from an individuals’ perception of their community food environment, the interpersonal relationships that impact one’s ability to navigate their food environment, a household’s financial stability, the availability of affordable housing, or systemic factors such as poverty, discrimination, and social inequities, we examine the interplay of individual and community-level factors influencing both objective and perceived food access among rural Latinos [10,11,12,13].

Most measures or definitions of food access consider the accessibility of sources of healthy food, as indicated by the distance to a store or the number of stores in an area, and household income or vehicle access [14,15]. However, a more comprehensive understanding recognizes that external factors beyond geography significantly influence a household’s ability to obtain food [15]. To fill this gap, researchers have adapted Penchansky and Thomas’s (1981) [16] classification of healthcare access dimensions: availability, accessibility, affordability, acceptability, and accommodation, to measure food access [11,17,18]. Availability refers to the adequacy of healthy food supplies, considering store types and their frequency, a quantitative variable. Accessibility examines the relationship between food sources and the target population, including geographic location, ease of access, and distance. Acceptability explores individuals’ attitudes and perceptions of their local food environment, as well as whether the available products meet their personal standards of quality. Affordability focuses on objective food prices and people’s perception of value relative to cost. Finally, accommodation assesses how well local food retailers adapt to residents’ needs, such as offering culturally relevant foods.

The five dimensions can also serve as a deliberate method to separate data collection. Studying food environments requires distinguishing between objective and perceived assessments, as each offers unique insights into how individuals interact with their surroundings [19,20]. The objective food environment refers to the measurable characteristics of the external food landscape, such as the number, type, location, and quality of food outlets, as well as the availability, price, and nutritional content of foods offered [21,22]. The strength of objective measures lies in their quantifiable aspect, enabling standardized comparisons among various populations and geographic areas. However, it may not fully capture the experiences and community settings of individuals that shape dietary choices. Assessments of perceived food environments incorporate individuals’ awareness, beliefs, and attitudes about their surroundings, including their perception of accessibility, affordability, or the quality and cultural appropriateness of available foods [19,23]. The strength of perceived measures lies in their ability to reflect personal experiences and barriers, which are critical for understanding individual dietary behaviors. However, they are susceptible to individual biases, recall errors, and may not always align with objectively measured characteristics of the environment [19,20]. Ultimately, to capture the complex interplay between the food environments and individual experiences, a comprehensive understanding of food access and dietary behavior necessitates the integration of both objective and perceived measures [10,24,25].

One way to involve community members in knowledge generation is through a Participatory Geographic Information System (PGIS) approach, also known as community GIS, qualitative GIS, or critical GIS. PGIS empowers community members to collect, analyze, and interpret spatial data [26], acknowledging the importance of local knowledge, lived experiences, and diverse perspectives in understanding spatial phenomena [27]. This approach provides researchers with community insight, as a way for data verification and an opportunity to apply context to datasets [26]. Very few studies in public health have engaged rural Latinos as the central focus of their design.

This study bridges the knowledge gap by exploring perceptions of food access among Latinos in rural areas, addressing the questions: “How do rural Latinos access food?” and “What challenges and resources do rural Latinos encounter when feeding their families?” It aims to understand the phenomenology of food access through mapping activities and semi-structured interviews that examine the subjective realities and lived experiences of this population.

## 2. Materials and Methods

### 2.1. Study Design

This phenomenological qualitative study [28] was conducted in rural communities in Nebraska, US, using a combination of participatory mapping, semi-structured interviews, and a demographic survey. In this study, PGIS principles were used to collect information on store availability and accessibility, enhancing the study’s dataset of store locations. Qualitative, semi-structured interviews provided insight into individuals’ experiences with their food environment [25]. Lastly, information on participants’ households, experience with food insecurity, and participation in federal benefit programs was collected through demographic surveys. Figure 1 outlines the sequence of activities conducted for all participants.

The methodology follows the Consolidated Criteria for Reporting Qualitative Research (COREQ), a 32-item checklist for interviews [29]. This study was approved in April 2024 by the University of Nebraska Medical Center Institutional Review Board (IRB# 0182-24-EX).

### 2.2. Participants and Setting

Participant recruitment and one-on-one interviews took place during the summer of 2024. Eligible participants were adults self-identifying as Latino, currently residing in rural areas with high Latino populations, and areas deemed as vulnerable to food insecurity [30]. The US Department of Agriculture (USDA) defines food-insecure households as those uncertain of having, or unable to acquire, enough food to meet the needs of all their members because they have insufficient money or other resources for food [31]. Exclusion criteria consisted solely of cases where another member of the same household had already participated in the study. The study aimed to recruit 18–24 participants, considering the project timeline and funding for travel and participant incentives. This study used three recruitment strategies. Initial recruitment was facilitated by local Community Health Workers (CHWs), who had established relationships with the research team and the rural Latino community through a previous study. Subsequent recruitment utilized Respondent-Driven Sampling (RDS), a network-based method similar to snowball sampling and word-of-mouth [32]. RDS is an advantageous sampling strategy for identifying hard-to-reach populations [33]. RDS proved effective in building trust between the research team and Latino participants, as recruitment was conducted through familiar acquaintances who had participated in the interview process and could vouch for its validity. In areas without prior relationships with CHWs, the research team contacted local organizations serving low-income Latino families, which then assisted with recruitment. Recruitment took place via text messages to participants from a previous study and in person through RDS and local organizations. Specifically, five participants were recruited from previous research groups, six through RDS, and seven through local organizations. Six individuals who were recruited did not attend their scheduled interviews due to scheduling errors (n = 2), childcare issues (n = 1), and no-shows (n = 3). Of the 24 invited participants, 18 completed the interviews.

It is worth acknowledging the influence of self-selection bias on the final sample. By consciously choosing to engage a population aware of food access challenges and leveraging established networks (CHWs and local partner organizations), this approach may have introduced a bias toward individuals who are more engaged, have more established connections, or who already utilize robust access to social services. This was a necessary trade-off to ensure the inclusion of participants with deep, lived knowledge regarding food access vulnerabilities, but it should be considered when interpreting the generalizability of the findings.

Interviews were conducted in-person at various locations, including public libraries, a university conference room, a nonprofit office, and local health clinics. Location selection prioritized free spaces, participant accessibility, and quiet environments with limited distractions. One in-home interview was conducted due to the participant’s lack of transportation and physical immobility, and appropriate safety measures were taken for the researcher. The research team opted for narrative consent since it is a common method with vulnerable populations [34]. Each participant received a $25 gift card as compensation for their time.

### 2.3. Data Collection

Data collection and analysis were led by the first author, who possesses extensive training and expertise in qualitative interviewing methods. Her linguistic and cultural competency fostered trust and open communication in the participants’ preferred language, English or Spanish, ensuring data quality and authenticity and minimizing misinterpretations during collection, transcription, and bilingual co-analysis. She developed the interview guide, conducted all interviews, and co-analyzed the resulting data.

Data collection was completed simultaneously through three different steps: mapping, semi-structured interviews, and a demographic survey. The interview script and a supplementary survey were developed collaboratively by the lead researcher and colleagues. The interview protocol, along with maps, a script, and a survey, was pilot-tested with other Spanish-speaking Latinas within the researcher’s department. Feedback on the maps included removing the legend box for the pink and blue dots indicating food retailers’ locations, removing citations to make the maps less intimidating, and using larger print and fonts. The script was edited for clarity and content based on the theoretical framework. Lastly, the only edit on the demographic survey was to add options on why participants do not participate in federal assistance programs.

#### 2.3.1. Step 1: Maps

Participants were initially asked about the availability and accessibility of stores through a facilitated sketch mapping activity [35]. This activity served to verify existing store location data and provided a visual representation of participants’ food environments, including their homes, workplaces, children’s schools, and shopping locations. This assisted in shifting participants’ mindsets, encouraging them to consider their relationship with food as a spatial and community-based experience. Furthermore, the mapping exercise allowed for the quantification of availability and accessibility by recording the number of stores frequented, transportation modes, travel distances, and trip-chaining patterns [36]. One of the most significant barriers to applying participatory mapping strategies is inequalities in access to the latest technologies and high-speed internet, known as digital divides [37,38]. Therefore, maps were created using ArcGIS Pro software version 3.1.3 and printed on large paper to facilitate easier use and reading for participants. Printing physical copies of maps not only removes the challenges of technology but also allows participants to draw, take notes, and move the map according to how they navigate their space daily, rather than using a north-facing map [36]. Figure 2 includes an example of the physical map used. The researcher also made an online map available to help participants identify locations, find addresses for items not initially included in the map, and assist participants as needed. Participants were asked to draw on the map where they live and identify stores where they shop by either marking stores already on the map or adding new stores that were not previously captured. We then discussed their routes to these stores and how activities such as school pick-up or drop-off, commuting to work, or transportation plans influenced their store choices.

#### 2.3.2. Step 2: Semi-Structured Interviews

The semi-structured interviews were based on the theoretical framework that food access comprises five dimensions: Availability, Accessibility, Affordability, Accommodation, and Acceptability (Table 1) [16].

The interviewer used a script with 11 questions, which were primarily open-ended and framed with neutral or positive language, avoiding jargon or terminology that might bias participants toward negative experiences of food insecurity (Table 2). Participants spoke in detail about the information on their map, their experiences with food access in their region, the availability of culturally relevant foods, and the challenges and resources in their communities. Interviews were recorded and conducted in the participants’ preferred language (English or Spanish).

#### 2.3.3. Step 3: Demographic Survey

Once the interview reached a stopping point, participants were instructed to complete the demographic survey. Participants were able to complete the survey by themselves or with the assistance of the interviewer, in their language of preference. The survey included questions adapted from the 2013 revised version of the US Household Food Security Survey Module in both English and Spanish, created by the Economic Research Service, USDA [39]. Use of federal benefit programs such as the Supplemental Nutrition Assistance Program (SNAP), also referred to as the Electronic Benefit Transfer (EBT) card, Additional questions about participants’ household demographics were adapted from the Perceived Nutrition Environment Measures Survey (NEMS-P) [40]. Variables collected information on household characteristics, participation in federal assistance programs, and three food insecurity screener questions.

### 2.4. Data Analysis

The analysis used methodological triangulation, systematically combining data from semi-structured interviews (qualitative), PGIS mapping (spatial data), and the study’s original store database [41]. This approach helps validate our findings by providing objective context and verifying the physical environment through the confirmation of store types and locations, supporting participants’ perceptions.

### 2.5. Maps

This study applied PGIS principles to gather information on spatial attributes and behavior patterns [42], in addition to validating and improving our database on local food retailers [43]. We predominantly focused on the Explore phase of the Explore, Explain, and Predict Model [42]. The Explore phase in PGIS utilizes GIS tools to identify patterns and assess data quality. It includes essential data cleaning and preparation steps, such as error correction, outlier removal, and reclassification. For the purposes of this study, the Explore phase assisted in answering research questions by providing context to the data collected during the interviews and validating the dataset.

Results from sketch mapping elicited participants’ perceptions and understanding of their environment by labeling or drawing on a reference map. Sites were classified as chain grocery stores (supermarket chains available in multiple regional locations), local grocery stores (full-service grocery stores found only in one municipality), ethnic stores (independently owned markets catering to an international audience), and other food retailers such as farmers markets, gas stations, and specialty stores (bakeries, butcher shops, tortillerias, etc.).

### 2.6. Semi-Structured Interviews

Thematic analysis applied categories from the interview codebook, which was developed a priori with pre-defined codes based on the five dimensions of food access [16]. A combination of a priori and emergent coding was employed using the five key dimensions: availability, accessibility, affordability, acceptability, and accommodation. While initial codes were developed based on the research framework to encompass themes from micro (individual and interpersonal) and macro (organization, community, and policy) levels, a “general comments” category was initially included to capture unexpected themes. During the analysis of each transcript, we utilized this category for open-ended questions, reading the text to identify novel codes relevant to the study that extended beyond the five ‘A’s of food access. We simultaneously took analytical notes on the significance of these new codes, which were then grouped by meaning to form emergent analysis categories [44]. Given the heterogeneity of participants’ experiences and the varied contexts of food access, thematic saturation was not anticipated. Instead, a target sample size of 18 to 24 participants was deemed appropriate, considering the number of research sites, available resources, and the project timeline. Interview transcripts were not returned to participants for review or feedback. The data cleaning process involved removing identifiable information, such as participants’ names, addresses, occupations, and other identifiers, while also using pseudonyms in the results section.

Interviews were recorded and transcribed verbatim in their original language. Two trained bilingual research team coders double-coded transcripts in their original language in NVivo. Inter-coder reliability was assessed using Cohen’s kappa (goal > 0.80). Resulting descriptive categories were compiled in English and Spanish and compared across languages until bilingual coders reached an agreement. Spanish transcripts were analyzed and coded in their original language, while English translations were completed solely for publication purposes [45]. Translation of participants’ quotes was performed by an English-Spanish bilingual researcher familiar with the cultural context, and edits were made to achieve conceptual equivalence of the quotes’ meaning across both languages, in order to preserve the authenticity and accuracy of what was said [45,46]. The transcriptions in their original language are available for consultation with the research team upon reasonable request. The descriptive data from the demographic survey were analyzed using SPSS software version 30.0.

## 3. Results

### 3.1. Participant Characteristics

In-person interviews (n = 18) were conducted with Latinas residing in eight rural towns, lasting approximately one hour. Eleven interviews (61%) were conducted in Spanish, and seven (39%) in English. The majority of women were married (61.1%) or living as a couple (5.6%). A significant proportion of participants were not employed (50%) (not seeking employment) or unemployed (11.1%). Education levels varied, with 38.9% of participants holding a college degree. The mean age of the participants was 38.06 years (SD = 10.1). Households had an average of 1.89 adults (SD = 0.9) and 2.28 children (SD = 1.5). Most households (72.2%) had an annual income between $20,001 and $50,000.

A substantial proportion of households reported experiencing food insecurity, with only 27% of participants reporting never experiencing difficulties with food access. 72.2% of participants reported often or sometimes worrying that food would run out before they had money to buy more. Similarly, 66.7% reported often or sometimes experiencing situations where food did not last and they lacked the funds to purchase additional groceries. Lastly, 66.7% of households indicated that they often or sometimes could not afford to eat a variety of nutritious meals. According to the USDA’s Guide to Measuring Household Food Security, the combination of these affirmative statements places 68.3% of participants in a “food insecure” status. Meanwhile, participation in government assistance programs was relatively low, with 61% of participants not eligible for benefits. Of those eligible, 27.8% of households received SNAP benefits, and 27.8% participated in the Special Supplemental Nutrition Program for Women, Infants, and Children (WIC) (Table 3).

### 3.2. Interview Themes

The interview themes presented were identified through thematic analysis conducted after the initial coding process was complete [47]. To reflect on participants’ experience with food access, a priori themes were based on the constructs of the five key dimensions of access, with the addition of emergent themes such as micro- and macro-level factors that appeared during the interviews To deidentify stores, we replaced store names with their designation of chain stores (CS), local stores (LS), ethnic stores (ES), or other stores (OS). Participants’ quotes were translated but left unedited to reflect their original voice and word choices.

#### 3.2.1. Availability

Table 4 summarizes the total count of identified food retailers by type and the number of participants who reported shopping at those locations, including those newly identified during the participatory mapping activity. Participants identified sixteen new stores that were not originally included in the study dataset. Participants mostly shopped at large chain stores (n = 18), despite the presence of ethnic stores in town. Some unique places where participants shopped for food included dollar stores (n = 6), farmers’ markets (n = 2), and specialty stores such as bakeries and butcher shops (n = 5). All participants mentioned chain stores as their primary shopping location, whereas ethnic stores were used for specific items or special occasions.

Most participants felt that there were stores available overall, that they had enough food options, and that healthy foods were available. The average population of the towns where interviews took place was slightly over 21,000, and the smallest town had a population of slightly over 8000. Every town had at least two grocery stores (chain or local), at least one ethnic store, and most participants were unconcerned about traveling out of town to buy food.

“it’s not like I often shop there [ethnic store] either, almost never. It’s only on special occasions that I go there to buy something. But the stores I usually frequent the most are [3 CS] and [2 dollar stores]” (Participant 5)

“Yeah, that is good. I like the stores that are running, they’re pretty good. Each one has their own…thing about them, you know. So Co-op’s my main go-to, but [I go to CS] Fridays for their sales, and [other CS] for whatever they don’t have, I go there.” (Participant 6)

“no, no, [ethnic store] is not my main shopping trip because we only go for pan dulce once a week… My main stores are [3 CS]” (Participant 1)

Despite the perceived overall availability, challenges exist with the availability of specific food items, particularly nutritionally tailored options (such as lactose-free, gluten-free, and diabetic-friendly foods). Participants also reported traveling out of town for major grocery trips, highlighting that stores are often located in central areas rather than distributed throughout town.

“I mean [city] is growing so…we definitely need more stores because they’re all kind of like right here, like in the middle.” (Participant 8)

“… there’s nothing there but a gas station and a dollar store. And, well, there’s nothing there, there’s nothing there that tastes good.” (Participant 10)

“Well, if you get to talk to [store], tell them they need lactose-free milk.” (Participant 4)

#### 3.2.2. Accessibility

Accessibility is hampered by the need for multiple trips to different stores to find specific items or stay within budget, which incurs costs like gas and time. A significant barrier highlighted is the lack of public transportation, which forces reliance on carpooling or personal vehicles, becoming problematic for recent migrants or elderly individuals. The lack of walkable spaces in rural towns is also an issue, with a participant citing pedestrian deaths as a stressor.

“Sometimes many people do what I’ve seen is [drive to multiple stores searching for cheaper prices]… I don’t like going from place to place… I don’t like to spend gas all the way to [CS] for a packet of cilantro.” (Participant 5)

“I feel bad for everybody else… I mean, unless you like call an Uber, but like somebody’s 60 year old Mexican grandma is not going to call Uber to take them to wherever you know or like Uber eats or something like, you know. So there’s like no public transportation.” (Participant 3)

“Yeah, a lot of it’s like carpool. You know, that’s it being an issue is because if someone has to go home and we ask them right away, do you carpool? And they’re like, “yeah”, so they can’t leave unless they’re carpool. It’s the same thing with the stores, “hey, when are you going to the store? Well, I’m not going for another two days” (Participant 2)

“…once a woman died years ago… she was walking. And about two years ago, someone else died… that was around 5:00 in the morning, the man ran her over, and it was a little dark. I don’t know why, but it does [worry] me a little bit.” (Participant 6)

#### 3.2.3. Acceptability

Ethnic stores are valued for offering authentic and culturally specific foods (e.g., specific avocados, chiles, plantains, cheeses), which are perceived as higher quality or better tasting than similar items in chain stores. However, concerns about quality control in ethnic stores were also mentioned.

“And if there’s something like avocados, [avocados from chain stores are] not as good as [ethnic store]… Like seasoned meat. If my kids are feeling fancy, we grab it from [ethnic store] too.” (Participant 4)

“I think chiles and pepitas, they do sell them at [CS], but I don’t know why, like it’s just different. They’re better from the Mexican store.” (Participant 8)

“And then sometimes the products are not even like up to quality. Because one time, I went to [ethnic store] to buy chorizo, and that thing was expired, and I didn’t even check it. So, I was like, “Oh my God…oh, this tastes funky” … then I looked up expiration date and it had expired 3 days ago… I literally spent six bucks on this chorizo for no reason… Never again, yeah. That’s why I don’t go to those stores anymore” (Participant 11)

#### 3.2.4. Accommodation

Participants reported that chain stores have increased their stocking of Hispanic products over time, improving accommodation. Yet, participants still noted a desire for more variety in culturally rooted, healthy cooking ingredients that are also affordable.

“When I first came here [in 2000], you couldn’t find beans, you couldn’t find tortillas, the pumpkin seeds, the ones used to make tamales, my banana leaf. Nothing… I remember maybe in 2005 or 2006, none of that was seen. But now, thank God, it’s over. Everything, everything, everything, everything. It’s easy now… Yes, now [CS] and [LS] carry more products, and it’s nothing like before.” (Participant 2)

“I think in [city] we definitely need more variety of stores like, mas tiendas. [CS] has the essential foods for every meal that you want to prepare. But if you want to stay cooking in your roots and your healthy way—we don’t have resources in [city] to buy our foods like that unless we go to the Mexican store, but everything is so expensive.” (Participant 9)

“When we go to [city], there are one or two Mexican stores that are very well stocked, they are nothing like the ones here, so we take advantage of the opportunity to visit that Mexican store in [city]. And well, there’s a [CS] [here], but since there are more Hispanic people there, you can find more products at [their] [CS], like the local tortilla brand… In fact, lately, there’s been a lot, a lot of Latinos [working in the store], and that’s what helps you feel more confident.” (Participant 13)

#### 3.2.5. Affordability

There was a mixed perception of food prices between large chain stores and local ethnic stores. Some found ethnic stores to be more expensive for general groceries, while others found specific cultural items or fresh produce to be cheaper there than at large chain stores. A major concern is the overall higher cost of food in rural areas compared to larger cities. Participants also noted higher prices for Hispanic-specific items, citing a mix of reasons, including an unbalanced supply and demand, store monopolies, and overall increasing prices.

“Honestly, I don’t [shop at the Mexican store] because I feel like their prices are more expensive than they are at just regular stores.” (Participant 11)

“Yes, because I like it, because I have to keep an eye on where it’s best for me. Because [at the Mexican store, squash] is about $1 a pound. And if I buy at [CS], it’s more expensive. Yeah, so that’s why I go [to the Mexican store] and I like papaya, so can I get papaya there, and vegetables. I also like to buy lemons because they’re not as fresh, [but it’s good enough for the price].” (Participant 6)

“I like [CS], but when it doesn’t have those things, I have to go [to ethnic store] because I need them. So even if I pay more, I’ll still get those things.” (Participant 14)

#### 3.2.6. Micro-Level Factors

Individual factors such as the desire to cook at home to reconnect with cultural roots, mental health challenges affecting work capacity, and personal resourcefulness influenced food access strategies. Participants often brought up the mental toll of navigating food choices, managing incomes, and the importance of holding onto their heritage through food.

“So for me, it’s super important to just cook at home, and it’s like ‘oh I’m cooking like my mom’s cooking’, you know, it reminds me of home. Or it reminds me when I was little, and it’s comforting. For me that’s important… even though I’m here now, and I grew up here, I went to school and everything. But it’s still, I’m still home when I eat those foods. That’s important.” (Participant 9)

“I got on Social Security because I had a mental breakdown…with my daughter that’s diabetic…. I had two jobs at that time. And with taking time off traveling to [doctor appointments in other states] in the winter… I think I did overwhelm myself and it just… killed my brain, I guess.” (Participant 16)

“I’m fine because I grew up in Mexico. I know how to live with little… The only thing is my illness. But we’re…fine… We know how to save, we know how to take care of others, there are no vices. No, I don’t even go to bingo anymore…” (Participant 18)

Participants often relied on word-of-mouth and community networks (e.g., school lunch programs, church pastors, carpooling) to access food, learn about resources, and overcome transportation barriers, as formal information channels were not always effective (e.g., they were unaware of benefits until a teacher informed them).

“Well, I don’t think [people know about those nutrition assistance programs]. I didn’t know about the card until it arrived at my house. And in fact, it arrived, and it took me about 23 months to find out what it was until a teacher told me,”(Participant 13)

“Almost all the Hispanic people I know use the free school lunch program.” (Participant 1)

“Word of mouth, [social media], sometimes pastors will go out and they’ll talk to the people. Like this pastor that we used to have. He would actually go out, drive around, and just try to reach people and just invite them to the church’s [food pantry].” (Participant 16)

#### 3.2.7. Macro-Level Factors

Community organizations played a significant role in food access. While food pantries and donations were acknowledged as helpful, there were significant concerns about the quality and type of food provided. It is not uncommon for food donations to consist mostly of canned and pre-packaged options, rather than fresh, healthy, or culturally appropriate choices. Participants also noted that they had, at times, received spoiled or expired items. One suggestion was for organizations to provide vouchers for grocery stores instead.

“I feel like there’s a lot of help out there in that regard for nutrition. It’s just like, take away your fear, your shame, and if you need it, just go find it.” (Participant 13)

“We never really qualify for food stamps when I came here. So, we kind of struggled and we would survive on food pantries… We would go to food pantries and it was good food. We would get vegetables, fruit, meat, chicken. You know good for the month—not like for the whole month—but good to give us some food in the meantime. But now… [the food pantries] are not very good like they used to be in the past… I think it’s just after COVID everything kind of changed. So, a lot of things are more expensive now and there’s not a lot of resource.” (Participant 9)

“There is not a healthy option in [the food pantry], and also with the mobile pantries—bless their hearts—because they’re helping us in the end, but it’s like a lot of canned items and prepackaged stuff or processed stuff… There are some that have some fresh produce in it, but not all. I wish there was more healthier options when people need assistance for food.” (Participant 4)

#### 3.2.8. Federal Assistance

Participants often mentioned school meals as part of their meal-planning considerations. Programs like the summer lunch program for children are highly valued for ensuring consistent meals.

“Last week they sent out school cards. That was a huge help for me, a huge help that you might not is enough… but that’s been enough for me for two weeks. When the pandemic started they started sending out the EBT cards to each parent, because the kids, depending on what they were eating at home, that was going to help the parents.” (Participant 2)

“They have a summer lunch program for kids. And that helps a lot because the kids get to eat lunch for free and I do know for some of the families that I’ve worked with they weren’t even sure that they would have dinner, but at least they had the peace of mind that they would have lunch… So, I really think that’s a great program, at least to make sure that our kids have lunch…” (Participant 4)

However, eligibility criteria for programs such as SNAP and housing assistance (Section 8) create significant barriers. Slight increases in income can lead to a loss of benefits, making it difficult for families to afford necessities like rent and utilities alongside food. This creates stress for families, to the point where a pay raise can feel like a pay cut, forcing people to consider undesirable trade-offs such as taking a lower-paying job, or getting a second job at the expense of losing family time, a hard decision, particularly for single parents or those struggling with childcare.

“So I recently got a raise at work. That wiped all my benefits. I was getting SNAP for my kiddos and that was a big blessing. As soon as I got my raise, it was more of a pay cut because all my benefits were wiped away… And when I got the last phone call, they told me my income was $2.18 over the limit and that’s why they wiped away all my benefits… You need a roof over your kids’ heads, so I wish they would take all that into account when they see their standards and they’re like ‘oh, you make too much’.” (Participant 4)

“[My daughter] applied for Section 8. They saw [her paycheck], and they said that she earned a lot… so she doesn’t qualify anymore. A house came up for [rent] but [Section 8] didn’t take her because [the rent] was too much. But how is she going to pay for everything? And what happens to her saving her money to continue studying? Who knows if she’ll be able to continue studying this coming year.” (Participant 7)

“… I have a good paying job, but I still feel like it’s not enough, because even though it’s just me and my daughter, I don’t qualify for EBT, I don’t qualify for Medicaid. I don’t qualify for any type of assistance. And there’s just times where it does get rough. … I’ll stay at my job; probably get a second job, see if that’ll help. But then if I get a second job, I’m losing time with my daughter, and I don’t want to do that.” (Participant 11)

#### 3.2.9. General Themes

Documentation status is a critical factor impacting food access, affecting eligibility for assistance programs, job access, and a steady income, and influencing fear of government interaction or perceived safety when seeking help. Even immigrants with legal visas find themselves in bureaucratic red tape that limits their ability to work. To support their families, individuals sought unregulated activities, such as making and selling their own food, caretaking, or cleaning jobs.

“We have a student that is undocumented, or the family is undocumented… We refer them to places that they can go and apply for food pantries, they get food every month. Like we would tell them [about WIC and the food stamp application]. ‘Hey, there’s this [federal] program, but you unfortunately don’t qualify for it,’ but we do have to tell them about the different programs that there is, you know. So even if they don’t have papers, we still provide the information” (Participant 9)

“…But the problem is like us, Latinos, many of us don’t have papers, so to get a [job at the local employer] you need Social Security, so we don’t have that, and that’s the hardest thing for someone who would like to work, but doesn’t have Social Security… if I want to have a Social Security, I would have to apply to a lawyer. But lawyers also charge a lot of money…What I can do, when I’m able, is to go clean houses, but with people I know.” (Participant 6)

“Some wives I know are biologists, some are nutritionists, I’m a nurse… But that’s how they hold us in their hands, so we’re dependent. But it is difficult, and there are many, many women who can’t work professionally…I have certain benefits that I can acquire because I have [legal] immigration status. But for us to be able to work so we can do other things, which is really what we want, no… No, we can’t. We can’t have vehicles in our names… So, yes, of course. This is an impact that would benefit food… many things in the house, like clothing, food, things we [could have] if we had been working. But we can’t work.” (Participant 15)

The rising cost of living, including rent, groceries, gas, and utilities, along with a perceived lack of job opportunities, was a significant stressor, making it difficult for even those with “good-paying jobs” to afford basic necessities. Families often found themselves relying on multiple incomes, which may depend on seasonality, and living from paycheck to paycheck while juggling other bills.

“I feel like nobody’s hiring. I always hear people like, oh, “I need a job”. Like nobody’s hiring. So where do I get a job, you know? And we only have, like, so many places here to work without going out of town.” (Participant 3)

“Prices are going up so bad where you can’t afford to have just one person working right now. I just feel like everything went crazy, expensive and the jobs stayed like, the pay is not enough for how the prices went up… I’m in the process of trying to apply for WIC and food stamps. But right now, we’re just struggling.” (Participant 8)

“I feel overwhelmed. Because it’s just me, you know, I’m doing everything by myself. I’m trying to provide for my daughter and, get us ahead in life, and at least try to live comfortably. But sometimes when all the bills come up—especially like hospital bills… how do you guys expect me to pay this while I’m still paying for rent and for groceries and for gas and insurance?” (Participant 11)

## 4. Discussion

This study showed the multifaceted challenges rural Latinos in Nebraska faced in accessing nutritious and affordable food, providing insight into how these communities perceived and navigated their food environments. Addressing food insecurity requires a comprehensive approach that extends beyond geographic availability to food retailers to encompass the dimensions of accessibility, affordability, acceptability, and accommodation. The high prevalence of food insecurity among Hispanic households nationally (21.9% in 2023) is starkly reflected in this sample, where 68.3% of participants met the criteria for food insecurity. This disparity highlights systemic inequities that are further exacerbated in rural settings [31].

### 4.1. Availability

The presence of grocery stores in every town, often cited as a sign of adequate access, is undermined by their centralized physical distribution. This concentration, a legacy of city planning that often overlooks equitable access in rural development [8,10], forces residents, especially those in less populated or historically underserved areas, to undertake multiple, often costly, trips. This is particularly burdensome when seeking specific dietary items, such as health-related or culturally unique products, which directly increase gas costs and demand considerable time.

### 4.2. Accessibility

The issue of accessibility is severely compounded by a pronounced lack of public transportation, which is endemic to many rural areas [48]. This forces an inequitable reliance on personal vehicles or carpooling, placing recent migrants, elderly individuals, and those without consistent car access at an extreme disadvantage. The absence of walkable spaces suitable for all seasons further isolates these communities, raising concerns about pedestrian safety and limiting independent mobility [12,49]. This fragmented food environment, where the necessity of navigating disparate and often niche ethnic stores exists alongside large, centrally located chains, points to a deeper issue than mere geographic distance. It reflects a systemic failure in policy and planning to address the unique needs of concentrated, often lower-income, populations within rural settings—populations that may face similar disadvantages to those resulting from different mechanisms of exclusion [12,48]. This creates a critical barrier to food access, highlighting how current infrastructure and historical planning oversights perpetuate food insecurity, rather than alleviating it.

### 4.3. Acceptability

The acceptability of available food sources was deeply intertwined with cultural relevance [50]. Participants highly valued ethnic stores for offering authentic and culturally specific foods, such as particular types of avocados, chiles, plantains, and cheeses. This preference was not without its nuances, as some participants expressed concerns about quality control in ethnic stores. Studies on Latino food access that focus on retail settings are often conducted in small ethnic grocery stores called “tiendas” or “bodegas” [51,52,53]. However, larger chain stores seem to be adapting to changing demographics, increasing the availability of traditionally Hispanic food items [54].

### 4.4. Accommodation

This accommodation has been noted by participants, as seen by the number of participants in all eight towns who stated they primarily shop at those places [55,56,57]. Despite these improvements, there is a strong desire for a greater variety of culturally rooted, healthy cooking ingredients that are also affordable. This indicates that although basic accommodation has begun, there is still a significant gap in addressing the diverse and nuanced culinary needs of Latino households, which is essential for promoting healthier dietary patterns and maintaining cultural heritage through food.

### 4.5. Affordability

Affordability emerged as a primary and often crushing barrier for all participants. Despite common misconceptions, living in rural areas is far from inexpensive. Research in the US consistently reveals that rural residents often face a higher cost of living due to increased prices for essential goods and services, including basic needs like utilities, housing, and healthcare, all the while having limited access to them [58,59,60]. Food, especially diet-specific foods, is more expensive in rural areas, forcing residents to travel to bigger towns, spending a larger portion of their household income on groceries and gas [58]. Most participants’ households (72.2%) had an annual income between $20,001 and $50,000, illustrating their vulnerability to price fluctuations.

### 4.6. Micro-Level Factors

Information networks among rural Latinos are also inconsistent, where individuals rely on interpersonal relationships to receive information on resources. However, misinformation spreads quickly among Latinos, as they report higher trust in information from their friends and family than other sources [61]. Studies recommend the use of promotoras, culturally competent community health workers (CHWs), to help with local outreach efforts and dissemination of health information [8]. However, access to promotoras is often linked to access to local healthcare systems, which is a barrier to Latinos in rural communities.

Whether it is dependence on transportation or how information spreads within a community, interpersonal relationships play a crucial role in how individuals navigate their food environment [13,62]. Challenges with transportation impact one’s ability to work, complete daily errands such as picking up children from daycare or school, and accessing grocery stores [48]. Participants’ struggles with transportation are barriers caused by the lack of local prioritization of public transportation, immigration policies that prevent dependents from having a vehicle in their name, and the costs associated with car ownership (such as down payments, monthly payments, gas, insurance, and maintenance).

### 4.7. Macro-Level Factors

During periods of economic downturns, food banks and pantries across the US see a sharp increase in demand, filling an important gap in food security for families. In 2023, following the end of pandemic SNAP benefits, nearly two-thirds of food banks reported a spike in demand for assistance [63]. Food bank usage has continued to rise, driven by rising food costs, low or no income, and the high cost of rent or home ownership [64]. Unfortunately, the nutritional quality of foods provided by food banks and pantries is not always up to customer standards [65], as pantries are often run by volunteers, and most lack standardization of practices for receiving donated food items. Along with the inconsistent availability of fresh or healthy foods, food bank participants also experience racism, shame, and discrimination.

### 4.8. Federal Assistance

The design and implementation of federal nutrition assistance programs, such as SNAP and housing assistance (Section 8), often exacerbate rather than alleviate food insecurity among low-income families. The eligibility criteria for these programs create unfair incentives, where small increases in income can unexpectedly result in a loss of essential benefits. This phenomenon, the benefits cliff effect [66], generates immense stress, forcing families to decide between affording food, rent, utilities, or other basic necessities [67]. This structural flaw in social safety net policies penalizes financial gains, trapping families in a cycle of poverty instead of providing a pathway to economic independence. Participants expressed concerns that their eligibility criteria fail to account for the cost-of-living expenses. However, according to the USDA, several types of bills can be considered when calculating income, which can increase benefit amounts or create eligibility. These bills include rent or mortgage payments, utility bills (e.g., electricity, gas, and water), childcare costs, and some medical expenses [68]. This type of misinformation, supported by literature, shows how difficult it may be to navigate assistance programs, especially with limited interpersonal relationships, mistrust [69,70], and systemic barriers such as limited digital access and familiarity with technology [37], language [47], and fear of advocating for themselves [71].

### 4.9. General Themes

Lastly, documentation status was a pervasive issue brought up by participants in most cities. The intersection of documentation status, visas, and immigration policies significantly impacts food and economic security for Latino families and primary caretakers [50,69,72]. The recent revocation of Temporary Protected Status (TPS) for individuals from Central America and the Caribbean has worsened these challenges [73]. Families are now vulnerable to deportation and lack legal work authorization, heightening their dependence on community resources [70]. Individuals under the dependent visa status face legal limitations that undermine household food security, due to policy-imposed restrictions on their ability to work legally or access crucial safety nets [74]. In response to these systemic limitations, individuals are often driven into precarious, unregulated economic activities, such as informal food sales, caretaking, or cleaning jobs, to meet their families’ fundamental needs.

Food insecurities lead to health concerns across the lifespan [75,76,77,78]. This study reminds us that, since this is not a new problem, improving health outcomes for Latino families requires a collaborative effort among local businesses, government leaders, healthcare providers, social services, and Latino families [69]. Comprehensive policy reform at the federal, state, and local levels is necessary to reassess immigrant eligibility for assistance programs, enhance food access, and promote economic independence for all U.S. residents, regardless of their immigration status.

### 4.10. Strengths & Limitations

This study collected diverse perspectives from Latinas living in areas vulnerable to food insecurity. A particular strength of this study was that it conducted interviews with participants before the recent rise in unease and fear related to immigration status. This proactive engagement allowed open conversations about documentation and immigration status, helping us build trust and better understand the unique challenges community members face.

However, the experiences of rural Latina women are not a monolith. A limitation of this study was the large size of the areas of interest. Different rural areas throughout the state vary in proximity to larger cities and community resources, as well as in acculturation levels and expectations regarding the availability of Latino foods. Therefore, reaching saturation among all interviews was unlikely, and data collection was stopped due to practical considerations, such as time constraints and funding for participant incentives, limiting generalizability beyond Nebraska’s rural contexts.

Lastly, we must acknowledge the influence of self-selection and gender bias on the sample. As detailed in the Methods Section 2.2, our reliance on established community networks was essential for recruitment. Still, it may have biased the sample toward individuals with robust social connections and prior awareness of social services. Our exclusive female sample may also be subject to bias. This gender imbalance may be attributed to women being more directly connected with our recruiters or having greater flexibility in their schedules to participate in the interviews, despite our efforts to be flexible with scheduling. This limits the generalizability of our findings and excludes the experiences of Latino men in similar contexts.

### 4.11. Future Work

Future research should integrate both objective and perceived measures of food environments, while leveraging community-led participatory methods to address the nuanced experiences of rural communities. Specifically, grants and policy research should focus on the pervasive structural barriers identified in this study that leave households exceptionally vulnerable to shifts in the political and economic landscape. Furthermore, studies are needed to quantify and address the economic burden of the “benefits cliff” effect by testing and evaluating policy recommendations, such as incorporating regional cost-of-living adjustments into federal assistance eligibility. Finally, events such as immigration enforcement actions, threats of government shutdowns affecting SNAP benefits, or the reduction in public health research funding can heavily and suddenly erode participants’ social stability and well-being. Future efforts must examine how state and local organizations can establish targeted, resilient, and non-federal assistance programs to provide stable food and economic support to individuals facing systemic barriers.

## 5. Conclusions

Our study highlights that food insecurity among rural Latinos is a complex issue driven by the intricate interplay of availability, accessibility, affordability, acceptability, and accommodation, all of which are framed by broader social determinants of health. The study design, which focused on objective and perceived data through multiple activities, was crucial to the quality of information gained. While interviews were central, PGIS also shaped how participants framed access to and use of space, providing insight into how they navigate and wish for their food environments. The findings highlight the need for interventions that go beyond increasing the number of stores. These interventions should focus on economic development and city planning, advocating for culturally relevant and affordable food options, and reforming assistance program eligibility and education without penalizing families.

## Figures and Tables

**Figure 1 ijerph-22-01800-f001:**
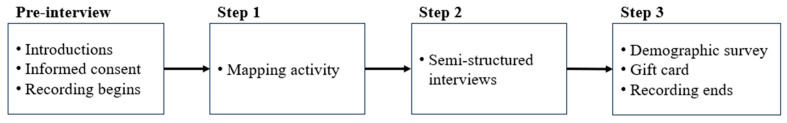
Data collection process for all participants.

**Figure 2 ijerph-22-01800-f002:**
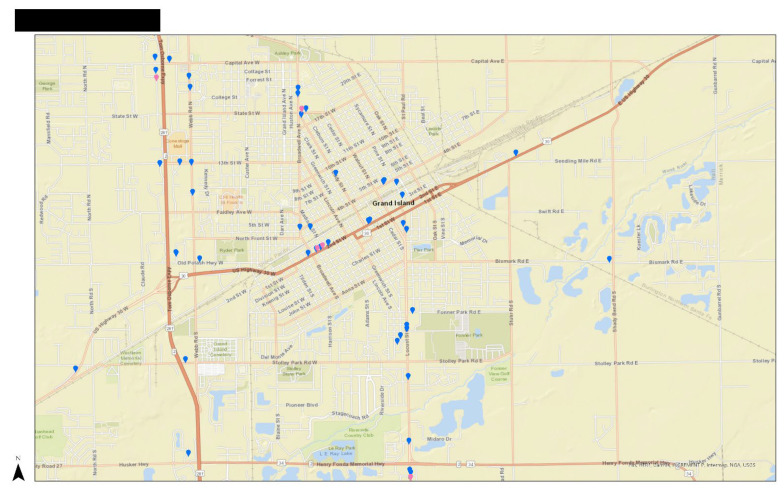
Sample physical map used with participants. The city’s name has been redacted.

**Table 1 ijerph-22-01800-t001:** Five dimensions of food access, adapted from Penchansky and Thomas, 1981 [16].

Dimensions	Definition	Assessment Tool	Level of Influence
Availability	Adequacy of the supply of healthy food (i.e., Store types, frequency)	PGIS	Community
Accessibility	Relationship between the point of interest and the target audience (i.e., Geographic location, ease of access, distance)	PGIS	Community Individual
Affordability	Food prices and people’s perceptions of worth relative to the cost (i.e., food prices, regional price indices)	Interviews	Consumer
Accommodation	How well local food sources accept and adapt to local residents’ needs (i.e., selling culturally relevant foods based on clientele)	Interviews	Community Consumer
Acceptability	People’s attitudes about the attributes of their local food environment, and whether or not the given supply of products meets their personal standards	Interviews	Individual

**Table 2 ijerph-22-01800-t002:** Semi-structured interview questions and probes.

Q#	Questions and Probes
Q1	I created a map of the area that you live in. The dots that you see are grocery stores that I found online and stores that other people me about. Can you find your home on this map? (collect address)
Q2	Do you shop in any of the stores that are on this map? Can you draw how you get to these stores from your house or work?
Q3	Are there any other stores that you shop at that are not included in this map? Can you draw where they are, or you can tell me and we can look it up together.
Q4	Can you tell me more about these stores? Why do you shop at these places/When do you go to one store instead of another? How often do you shop there? How does the price or cost differ between these stores? What foods do they have available?
Q5	Can you tell me more about how you get to these stores? How long does it take for you to get to the store? How easy is it to get there? Do you go with friends or family?
Q6	Is there a favorite or traditional dish you like to prepare? Where do you find ingredients to make that?
Q7	Would you change anything about these stores or your community and how you are able to buy foods? Like I mentioned in the beginning, I’m trying to understand more about how Latinos in rural areas access food. I’ve asked about places where you shop and how you get there and what you think about their products.
Q8	What do you think are the challenges that families or you face in feeding your families?
Q9	How do you or other families get help to feed their families? Do you know of any other places where people can get help with accessing food? (add to the map) (Provide examples if they need it) community garden, church food pantry, school breakfast and lunch programs or after school programs?
Q10	Could you suggest anyone else that might be able to help me with my study and answer these questions?
Q11	Do you have any questions about what we talked about today the activities that we did or what this study is about?

**Table 3 ijerph-22-01800-t003:** Participant Demographics (N = 18).

Variables	n	%
**Marital Status**		
Married	11	61.1
Living as a couple	1	5.6
Divorced	2	11.1
Widower	1	5.6
Single, never married	3	16.7
**Employment Status**		
Full-time	6	33.3
Part-time	1	5.6
Unemployed *	2	11.1
Not employed *	9	50
**Education**		
8th grade or less	4	22.2
Grades 9–12 (some high school)	2	11.1
High school graduate or GED	1	5.6
Some college or technical school	4	22.2
College graduate	7	38.9
	**mean**	**(SD)**
**Age**	38.06	10.1
**Adults in the household**	1.89	0.9
**Children in the household**	2.28	1.5
**Vehicles in the household**	1.89	0.8
**Benefits ****	**n**	**%**
SNAP	5	27.8
WIC	5	27.8
Government Cash Assistance ***	2	11.1
NB ^†^: not eligible	6	33.3
NB ^†^: Eligible	4	22.2
NB ^†^: Don’t know	1	5.6
**In the last 12 months…**	**n**	**%**
**Worried food would run out**		
Often true	2	11.1
Sometimes true	11	61.1
Never true	5	27.8
**Food didn’t last, didn’t have money to buy more**		
Often true	2	11.1
Sometimes true	10	55.6
Never true	6	33.3
**Couldn’t afford to eat a variety of nutritious meals**		
Often true	2	11.1
Sometimes true	10	55.6
Never true	6	33.3
**Household yearly income**	**n**	**%**
$0–20,000	3	16.7
$20,001–50,000	13	72.2
$50,001–75,000	2	11.1

* Unemployed = actively seeking employment/Not employed = not seeking employment (i.e., student, retired, homemaker, disabled, etc.). ** Benefits do not add to 100% due to multiple-choice questions. *** Government cash assistance (i.e., TANF, SSI, SSDI, GA). ^†^ NB = No Benefits. The participant does not use benefits due to ineligibility or lack of knowledge.

**Table 4 ijerph-22-01800-t004:** Store identification and participant shopping choices.

	Chain Stores (CS)	Local Stores (LS)	Ethnic Stores (ES)	Other Stores (OS) *	Total
Store identification method					
Stores in the original study dataset	13	5	10	6	34
New stores identified via PGIS	0	3	7	6	16
Total unique stores (Dataset + New)	13	8	17	12	50
Participant shopping choices					
Participants reporting shopping in each store type	18	7	8	7	N/A **

* OS (Other Stores) includes dollar stores, farmers’ markets, bakeries, and butcher shops. ** N/A (Not Applicable): The sum of participants across store types exceeds the total number of participants (n = 18) because most participants shop at multiple store types.

## Data Availability

The data presented in this article are not readily available due to participant privacy and ethical restrictions. Requests to access de-identified datasets should be directed to the corresponding author.

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
