# Peer review of "(registering DOI)"

_ijerph, 2025, doi:10.3390/ijerph22121800_

Round 1
Reviewer 1 Report
Comments and Suggestions for Authors
Revisit especially the methodology, results and discussion as in the review comments to improve the paper.

Grammar check recommended
Author Response
Thank you for taking the time to review our manuscript. We appreciate the reviewers' thoughtful considerations and insights, which will help us improve our work. We carefully read all their comments and addressed their suggestions point by point as requested.
- We have ensured all references are relevant to the content of the manuscript.
- We are submitting two versions of the manuscript.
- One that includes track changes and highlights major revisions to the manuscript, so editors and reviewers can see any changes made. This was primarily done for grammar, rephrasing, and editing typos.
- The second version is a clean version of the changes made, with highlights, track changes, and any other review tools removed.
Please see our point-by-point responses in the attached document.
-Natalia Santos

Reviewer 2 Report
Comments and Suggestions for Authors
This manuscript is well-organized and methodologically robust, highlighting the critical issues of food insecurity, cultural factors, and structural inequities in rural Latino communities. The participatory approach and bilingual rigor are commendable.
To strengthen the manuscript, I recommend:
- Expanding on reflexivity and research positionality
- Clarifying the integration of emergent codes and thematic development
- Tightening the policy implications to more directly reflect your data
- Enhancing visual clarity of certain tables/figures
After these revisions, I believe the manuscript would make a strong contribution to the public health and food security literature.

Author Response
Thank you for taking the time to review our manuscript. We appreciate the reviewers' thoughtful considerations and insights, which will help us improve our work. We carefully read all their comments and addressed their suggestions point by point as requested.
- We have ensured all references are relevant to the content of the manuscript.
- We are submitting two versions of the manuscript.
- One that includes track changes and highlights major revisions to the manuscript, so editors and reviewers can see any changes made. This was primarily done for grammar, rephrasing, and editing typos.
- The second version is a clean version of the changes made, with highlights, track changes, and any other review tools removed.
Please see our point-by-point responses in the attached document

Reviewer 3 Report
Comments and Suggestions for Authors
The manuscript “Policy, Price, and Perception: A Phenomenological Qualitative Study of the Rural Food Environment Among Latina Households” addresses an important and timely topic concerning the rural food environment among Latina women. The study makes a valuable contribution to understanding barriers to food access in marginalized rural communities and highlights the lived experiences of Latina women. However, in my opinion, several revisions are still needed before the manuscript can be considered for publication.
The Introduction is clear and provides sufficient background and rationale for the study. It effectively outlines the problem and establishes the relevance of the topic. No major revisions appear necessary in this section.
The Methods section would benefit from a more detailed description of key procedures. The authors should include the date of ethical approval and provide a fuller account of how narrative consent was obtained. It might also be helpful to include one or two examples of the questions used to confirm participants’ understanding and consent, which would make the process more transparent to readers.
The sample size of 18–24 participants has been justified in the manuscript by referencing time constraints, which seems acceptable given the qualitative design and available resources. However, it would strengthen the Methods section if the authors could provide more specific information regarding the data collection process, particularly the average duration of each interview. Including this detail would clarify the scope of participant engagement and the practical considerations that influenced the study design. In addition, while the focus on women is understandable in light of the study’s aims, it might be useful to briefly discuss in the Methods section the rationale for including only female participants and the implications this has for the generalizability of the findings.
Some of these methodological constraints, such as the limited sample size and the inclusion of women only, are already acknowledged in the Limitations section, which clearly shows that the authors are aware of the study’s boundaries. Still, these aspects should not appear solely as limitations but also as intentional methodological decisions. It would therefore be preferable to describe them more clearly in the Methods section, where the reasoning behind these design choices can be properly explained.
The paragraph describing the first author’s background, bilingualism, and prior experience feels overly detailed for the main text. For example, it does not seem relevant to inform readers that the first author is a PhD candidate. This could be reformulated into a concise statement focusing on how the author’s background may have influenced the research process, data collection, and interpretation.
Table 3 also requires some reformatting. Presenting decimal values for discrete variables such as adults, children, and vehicles per household is somewhat confusing. These data would be clearer if grouped into meaningful categories (for example, none, one, or two or more).
The Discussion section would benefit from a broader and more integrative approach. The authors interpret the results well, but they should more systematically compare their findings with similar research conducted in other settings or among different populations. This comparison would strengthen the discussion and situate the study within the wider literature on food access and social determinants of nutrition.
Although the manuscript currently includes some elements of Future Perspectives in the Conclusions section, it would be more appropriate to place it at the end of the Discussion.
Author Response

(The authors gave the same response as above.)

Round 2
Reviewer 2 Report
Comments and Suggestions for Authors
Thank you for your thorough revisions. The manuscript is now well-organized and clearly addresses all prior comments. The added explanation of the phenomenological rationale, data analysis transparency, and the strengthened discussion of PGIS meaningfully improve the paper. I find the revised version clear, coherent, and ready for publication.
Author Response
|
Thank you for your thorough revisions. The manuscript is now well-organized and clearly addresses all prior comments. The added explanation of the phenomenological rationale, data analysis transparency, and the strengthened discussion of PGIS meaningfully improve the paper. I find the revised version clear, coherent, and ready for publication. |
Thank you for your detailed review and suggestions to make this a stronger manuscript. |
Reviewer 3 Report
Comments and Suggestions for Authors
I find the revised manuscript improved, and several of my previous comments have been appropriately addressed. However, comparing your results with previous similar studies would further improve the manuscript.
Author Response
Reviewer 3:
“The manuscript “Policy, Price, and Perception: A Phenomenological Qualitative Study of the Rural Food Environment Among Latina Households” addresses an important and timely topic concerning the rural food environment among Latina women. The study makes a valuable contribution to understanding barriers to food access in marginalized rural communities and highlights the lived experiences of Latina women. However, in my opinion, several revisions are still needed before the manuscript can be considered for publication.”
Author's response:
Dear reviewers,
Thank you for your comments on our figures and tables.
We submitted a separate file with a higher resolution for the sample map used with participants. This is the exact map that was used; however, the name of the city has been redacted to keep a certain level of privacy. If you believe this is unnecessary, we can send the original copy without the redacted city.
We have also standardized all tables, so they follow the same formatting. Please let us know if there are any specific edits or formatting beyond what was revised.
|
I find the revised manuscript improved, and several of my previous comments have been appropriately addressed. However, comparing your results with previous similar studies would further improve the manuscript. |
We have edited the discussion section to include more comparisons with studies on Latina’s food access, preferences, and challenges with food security. Thank you for encouraging us to improve our discussion. |